# Metabolic Profiles, Genetic Diversity, and Genome Size of Bulgarian Population of *Alkanna tinctoria*

**DOI:** 10.3390/plants12010111

**Published:** 2022-12-26

**Authors:** Milena Nikolova, Ina Aneva, Petar Zhelev, Ivanka Semerdjieva, Valtcho D. Zheljazkov, Vladimir Vladimirov, Stoyan Stoyanov, Strahil Berkov, Elina Yankova-Tsvetkova

**Affiliations:** 1Department of Plant and Fungal Diversity and Resources, Institute of Biodiversity and Ecosystem Research, Bulgarian Academy of Sciences, 1113 Sofia, Bulgaria; 2Department of Dendrology, University of Forestry, 1797 Sofia, Bulgaria; 3Department of Botany and Agrometeorology, Agricultural University, Mendeleev 12, 4000 Plovdiv, Bulgaria; 4Department of Crop and Soil Science, Oregon State University, 3050 SW Campus Way, 109 Crop Science, Building, Corvallis, OR 97331, USA; 5Botanical Garden, Bulgarian Academy of Sciences, 1000 Sofia, Bulgaria

**Keywords:** ononitol, phenolic acids, aerial parts, total alkannin content

## Abstract

*Alkanna tinctoria* (L.) Tausch Boraginaceae is a medicinal plant whose root is used for its antimicrobial and anti-inflammatory properties. *A. tinctoria* roots have been subject to numerous studies. However, the aerial parts have been explored less. The objective of the present study was to compare the chemical profile of aerial parts and roots as well as the total alkannin content in roots of 11 populations of the species from different floristic regions of Bulgaria. Methanolic extracts from 22 samples were analyzed by GC/MS. Phenolic, fatty, and organic acids, sterols, polyols, fatty alcohols, and sugars were identified. Ononitol (4-O-methyl-myo-inositol) was found as the main compound in the aerial parts. The total alkannin content in the roots was evaluated by the spectrophotometric method and compared with that of the commercial product. Populations with high alkannin content and rich in other bioactive compounds were identified. A relatively low genetic diversity in the studied populations was observed. The present study is the first comprehensive study on metabolite profiles and genetic diversity of the Bulgarian populations of *A. tinctoria*. The occurrence of ononitol in the aerial parts of the species is reported for the first time, as well as the phenolic acid profiles of the species in both aerial parts and roots. The results showed that aerial parts of the plant are also promising for use as a source of valuable biologically active substances.

## 1. Introduction

*Alkanna tinctoria* (L.) Tausch (alkanet, alkanna) is a medicinal plant native to Southern Europe, Northern Africa, and Southwestern Asia. The species has limited distribution in Bulgaria; its habitats are located only in three floristic regions of the country in fragmented populations. The roots of the species are used as a source of red pigments in traditional communities and as a dye in the cosmetics, food, and textile industries [1,2,3]. The roots are also used as a natural remedy to prevent and treat ulcers, wounds, fever, inflammation, aging, and herpes [1,4]. In Bulgarian folk medicine, *A. tinctoria* roots are applied internally in the form of a decoction as an astringent for gastrointestinal disorders, and externally, to prepare ointments for application to wounds, pustules, and burns, and poultices made from the whole plant are used to treat mumps.

As the root is the used plant part of *A. tinctoria*, the majority of studies have been focused on the chemical composition and biological activity of the plant roots [2,3,5,6,7,8]. Studies on the other plant parts are rarer. Phytochemical analysis on *A. tinctoria* leaves extracts revealed that aqueous and ethanol extracts contained biologically active compounds, representatives of main chemical groups such as alkaloids, bufadienoloides, carbohydrates, flavonoids, gallotannins, phenolics, proteins, pseudotannins, resins, saponins, steroids, tannins, triterpenoides but individual compounds have not been identified [8]. The main phytochemical components of *A. tinctoria* roots that are responsible for its biological activity are the naphthoquinones alkannin, shikonin, and their derivatives: acetylalkannin, angelylalkannin, 5-methoxyangenylalkannin, acetylshikonin, dimethylacryl alkannin, naphtharzin, arnebifuranone, shikalkin, alkanfuranol and alkandiol [2,3,5,6]. A number of analytical techniques have been applied for the determination of the alkannin/shikonin (A/S) content of *A. tinctoria* [9,10]. High-performance liquid chromatography (HPLC) has been the most frequently utilized technique for qualitative and quantitative determinations of A/S and their derivatives [11,12,13], spectrophotometric method for quantitative assay of A/S content has been also applied [14,15,16].

The genetic variation of the populations of any species is a fundamental characteristic determining its capacity to persist and adapt to environmental changes [17,18]. The investigation of genetic diversity and structure of endangered species is of main importance for genetic resource conservation and plant breeding programs [19,20]. Revealing the genetic diversity of natural populations and cultivars is a basic requirement for the development of sustainable agricultural practices [21].

The genetic diversity of *Alkanna* species has been studied by employing different genetic markers and for different purposes. Wolff et al. [22] applied RAPD markers in their study on *A. orientalis* in the Sinai Desert, Egypt, and concluded that the main drivers of gene flow are periodic flash floods. The same classes of markers were applied by Abdel-Hamid [23] to distinguish between two subspecies of *A. tinctoria* in Lybia. Recently, more informative codominant markers have been developed for describing the genetic diversity of *Alkanna* species [24]. Genetic variation of three endemic *Alkanna* species in Bulgaria was studied by Semerdjieva et al. [25] based on highly polymorphic ISSR markers. The authors provided important inferences about the taxonomic and evolutionary relationships of the studied species. Taxonomic markers, like rpl32-trnL (UAG) и trnH–psbA [26], were applied successfully to different systematic groups, including the species of the Boraginaceae family. Even though these markers have some limitations, they could bring additional insight into the phylogenetic relationships among the *Alkanna* species. The genetic structure and its determinants in *A. tinctoria* and *A. sieberi* from Greece were studied by Ahmad et al. [27].

The genome size is a highly relevant characteristic of living organisms that presents the DNA content of the non-replicated gametic chromosome set [28,29]. It is defined with the term C-value proposed by Swift [30]. This value has been considered characteristic and invariable for each species. Information on the C-value can be useful in different areas of plant science, including ecology and phytogeography [29,31,32]. A correlation between genome size and environmental conditions has been reported by Bennett and Smith [33] and Grime and Mowforth [29]. Relationships between DNA amount and latitude [33], altitude [34], temperature [35], fertilizer treatment [36,37], and seedling growth rate [38] have been found.

The citometrical studies in *Alkanna* species are few. The genome size of *A. tinctoria* and *A. sieberi* was previously reported by Ahmad et al. [27].

The aim of the present work was to compare GC/MS-based metabolite profiles of the aerial parts and roots of 11 populations of *A. tinctoria* from different floristic regions as well as total alkannin content in roots in correlation with their origin (genetic profile and genome size) in order to facilitate the selection of desirable accessions for possible cultivation. The introduction of *A. tinctoria* into culture would preserve its natural populations in the Bulgarian flora.

## 2. Results

### 2.1. Phytochemical Study

#### 2.1.1. GC/MS Analysis

Twenty-four (24) metabolites were identified in the methanolic extracts from 22 samples of roots and aerial parts of *A. tinctoria* following GC/MS analyses (Appendix A Table A1 and Table A2). The identified compounds were phenolic, fatty, and organic acids, sterols, polyols, fatty alcohols, and sugars (Appendix A Table A1 and Table A2). Similar metabolic profiles with the quantitative variation of individual components between the studied samples of the different populations were found. Sucrose was the main metabolite in the roots. Monosaccharides were identified in large amounts also. Caffeic acid was the most abundant phenolic acid. The highest amounts of phenolic acids were found in the samples from the Danubian Plain (AT1, AT2). Eastern Rhodopes (AT11) and the River Struma Valley (AT3, AT9).

Ononitol (4-O-methyl-myo-inositol) was identified as the main component in the aerial parts of the studied samples (Table 1 and Figure 1). Ononitol identification in the extracts was made by comparing its mass spectra and RI with mass spectral databases (NIST; GOLM; Figure 2). The highest content of ononitol was found in the samples from the River Struma valley (AT3, AT10). In addition, the samples from this location had the highest content of sugars and sugar derivatives. Seven phenolic acids were identified; among them, caffeic, 4-hydroxybenzoic, and 4-hydroxycinnamic were the main ones. The highest content of phenolic acids was found in the samples from the River Struma Valley (AT10, AT5, AT3, AT9). Whole phenolic acids were more abundant in the aerial parts than in the roots. Sterols and sucrose were present in higher quantities in the roots than in aerial parts.

Generalized comparative data of established qualitative and quantitative metabolite composition of plant parts (roots and aerial parts) of studied *A. tinctoria* populations are presented in Table 1. The aerial parts of the species contained 4-hydroxybenzoic and caffeic acids and all identified organic acids in larger quantities than in those in the roots. However, the roots contained higher concentrations of monosaccharides, disaccharide (sucrose), sterols, and fatty alcohols. Ononitol was identified only in the aerial parts of the species.

#### 2.1.2. Spectrophotometric Analysis of Alkannin Derivatives

The total alkannin content in the hexane extracts of the roots of the studied samples was determined spectrophotometrically (Table 2). The highest alkannin content, exceeding that of the control sample, was found in the root extract of the population from Kresna town (AT3) in the Struma Valley region. Comparable alkannin content to that of the control was found in the sample from Lebnitsa village (AT8), Ilindentsi village (AT4), Kulata village (AT10), and also from the River Struma Valley region as well as Ostrov village, Danubian Plain (AT3). The lowest alkannin content was found in the sample from Spatovo village, the Struma Valley region (AT8), and Odrintsi village, at the Eastern Rhodopes Mountains (AT11).

### 2.2. Study of Genetic Diversity

The percentage of polymorphic bands in the studied populations ranged from 55 to 72 (mean 61.5%) and was the highest in the northern population AT1 (Table 3). A similar percentage of polymorphic loci (64.58) was established previously in other *Alkanna* species from Bulgaria (*A. graeca*) by Semerdjieva et al. [25]. The genetic distances among populations were relatively low, they ranged from 0.01 to 0.14, and the most distinct was the population Orsoya (AT1), situated at 1–1.5 km distance from the Danube River and at an air distance of at least 200 km from the remaining populations (results not shown). The cluster dendrogram (Figure 3) demonstrates the distinct position of population AT1 while the other populations are close to each other, and the grouping does not indicate a particular trend of spatial variation.

The estimated genetic diversity level in the studied population of *A. tinctoria* was low (He = 0.220–0.298), and with similar values in individual populations (Table 3)

### 2.3. Flow Cytometry Analysis

The estimated values of the DNA content in fresh leaves of the studied specimens of *A. tinctoria* are presented in Table 4 and show remarkable congruence—1C = 1.326 pg in both studied populations.

The estimated values of the DNA content in the seeds of *A. tinctoria* are presented in Table 5.

The CV of measurement of 5000 PI-stained particles in the samples, each containing 10 mature seeds from 11 *A. tinctoria* populations (AT1 to AT11), with no internal standard, varied in the range 4.7–10.7%. No difference between the populations was detected in the position of the peaks on the obtained histograms, i.e., similar fluorescence mean values were estimated for the different samples under the same gain values of the flow cytometer.

## 3. Discussion

### 3.1. Phytochemical Study

#### 3.1.1. GC/MS Analysis

Primary and secondary metabolites were identified by GC/MS analysis in the aerial parts and roots of *A. tinctoria*. The identified lipid compounds in the roots of the species are in accordance with previously reported data [39]. The present study provided data on phenolic acids profiles of the species both in the aerial parts and roots for the first time. Previous studies concerning the chemical composition of aerial parts of *A. tinctoria* reported only total phenolic content without identification of the individual compounds [1,40]. An important result of the present study was the identification of ononitol in the aerial parts of *A. tinctoria*. Because chemical composition studies have previously been focused on *A. tinctoria* roots, ononitol has not been reported in this species until now. Ononitol (4-O-methyl-myo-inositol) belongs to the group of polyols or cyclitols (Inositols). This compound, together with pinitol, are distinguished from the other methyl derivatives (such as bornesitol) and isomers of inositol by the presence of fragment ion at *m*/*z* 260 in their mass spectra [41]. Previous studies reported that ononitol exhibited important biological activities such as hepatoprotective, analgesic, and anti-inflammatory [42,43]. It has been reported that O-methyl-inositols such as ononitol are accumulated in plants in response to abiotic stresses such as drought and salts [44,45]. Ononitol has been reported mainly for Fabaceae species but also for Brassicaceae and Ericaceae [46]. In Boraginaceae, bornesitol (1-O-methyl-myo-inositol) seems to be widespread [47], but information about the distribution of ononitol in this family was not found. The occurrence of ononitol in the aerial parts of *A. tinctoria* becomes the first report on the presence of ononitol in the family Boraginaceae.

#### 3.1.2. Spectrophotometric Analysis

Differences in roots alkannin content between samples collected from different floristic regions and from different localities within one region were found. The alkannin content found in this study was comparable to those reported previously on Bulgarian populations of *A. tinctoria* [14]. Genova et al. [14] studied roots of the species from five localities in the Struma Valley Region different from those of the present study, and found variation of alkannin content from 3.40% to 13.00%. The latter authors found the highest amount of alkannin in the corky tissue of the main and lateral roots of medium adult individuals (10.06–12.85%) as well as in the corky tissue of young roots (11.99%). Less alkannin content was found in the secondary cortex, and no alkannin was found in secondary wood [14]. Papageorgiou et al. [48] reported also that the root age was crucial for shikonin content of *Lithospermum erythrorhizon* Sieb. et Zucc. In the cell suspension cultures, it has been found that shikonin derivative production was strongly affected by light, temperature, and pH of the medium. Alkaline pH (7.25–9.50), temperature 25 °C, and dark were reported to be optimal conditions for increased alkannin production [16]. In the present study, a low alkannin content was found in the sample (AT11) that was collected from calcareous rock, which gives us reason to assume this rock is probably unfavorable to accumulation of alkannin

Overall, the different amounts of alkannin and other biologically active compounds synthesized in each of the studied populations of *A. tinctoria* could be due to a number of factors such as (1) genotype; (2) soil type; (3) exposition; (4) environmental conditions; (5) the content of the available soil macro and micronutrients [49,50,51]; (6) fungal infections [52,53,54,55]; and (7) physiological status of plants [56]. Environmental factors are often proved to have a significant effect on the concentration, with lower effect on the composition of biologically active substances. In the case of *Sinopodophyllum hexandrum*, climate factors were more important than the soil properties [57]; however, the majority of studies revealed that these peculiarities are species-dependent [58,59]. The studied populations in the Struma River Valley (AT6, AT7) and the Eastern Rhodopes Mountain (AT11) are influenced by a Mediterranean climate, but their habitats differ. For example, the species distributed in the Eastern Rhodopes Mountain grow in Leptosols soils or calcareous rock with low anthropogenic impact due to the area’s depopulation. In contrast, in the Struma River Valley, the species grow on the Siliceous or Calcareous stony slopes in grassy communities. The species in these populations (AT6, AT7) have been exposed to greater anthropogenic impacts such as grazing and trampling. The conditions in the populations of the Danubian Plain were different (AT1, AT2), where the species grow on Inland sandy dunes and are influenced by a temperate-continental climate (Table 6). The present results confirmed previous research that, in general, the concentrations and type of plant secondary molecules are determined by the species, genotype, developmental stage, physiology, and environmental factors during growth [56].

### 3.2. Study of Genetic Diversity

Studies on the distribution of genetic diversity in the populations of *Alkanna* species have shown that, in most cases, there were no noticeable differences in the level of intrapopulation diversity among populations. The assessment of three endemic *Alkanna* species in Bulgaria by ISSR markers revealed close values of the diversity parameters, such as percent of polymorphic loci, expected heterozygosity, and polymorphism index of Shannon [25]. The preliminary results of our study have shown that the level of genetic diversity in *A. tinctoria* was comparable with that of the other *Alkanna* species. It should be noted that the comparison in this study was based only on one type of marker (ISSR) with dominant inheritance. Including other types of markers, such as microsatellites, or simple sequence repeats (SSR), already developed for *A. tinctoria* [24] could provide valuable additional information about the level of genetic diversity within and among populations.

This study demonstrated a low level of genetic diversity in the Bulgarian population of *A. tinctoria* that correlates with its limited distribution. Lower genetic diversity is attributed to endemic species and to species that are not widely distributed [60]. The low level of genetic diversity in their populations is due to the non-plasticity caused by the narrow distribution [61]. In turn, the reduction in genetic variation might suggest a decline in adaptation to a changing environment, leading to an increased danger of extinction and increased inbreeding [62,63]. In the studied Bulgarian populations of *A. tinctoria*, the established genetic diversity was low, while in the Greek populations of the species, Ahmad et al. [27] found moderate genetic diversity. This fact is consistent with the assumption of the same authors that limitation in population size and distribution is a major factor influencing population genetic diversity, which decreases with decreasing population size: the species is more common in Greece and other Mediterranean countries compared to Bulgaria, where it is considered very rare [64].

### 3.3. Flow Cytometry Analysis

This is the first report on the DNA content in the Bulgarian accessions of *A. tinctoria*. The estimated value of 1C = 1.326 pg was congruent with the data published for the species from Greek accessions (1C = 1.21–1.28 pg, [27]). Studies in the genome size in mature seeds, 10 seeds per population in a total of 11 populations (AT1 to AT11), revealed that 1C varied in the range of 1.28 to 1.49 pg (Table 4). As it was observed in the Greek populations [27], no pattern of variation of the C-values was observed speaking in favor of variation in the ploidy level within and between populations. Moreover, the observed small variation did not exhibit any geographical pattern. For example, the range of variation of the two accessions from the Danubian Plain floristic region in North Bulgaria was 1C = 1.29–1.31 pg, whereas, in the Struma River Valley in the South Bulgaria region, the variation was 1C = 1.28–1.43 pg. The analysis of the measurement histograms of the samples, each containing 10 seeds of *A. tinctoria* without internal standard, showed there were no different classes of the observed fluorescence mean values (i.e., only one peak per run present). This suggests there is no variation in the ploidy level between the different seeds (respectively, different individuals). This finding agrees with the established normal running of the reproductive processes leading to the formation of reduced male (sperm cells) and female (embryo sacs) gametes in *A. tinctoria* [65].

In general, flow cytometry is a useful screening method for estimating the reproductive development pathways using mature seeds. Matzk et al. [66] presented 10 different pathways of seed formation depending on the participation of reduced or unreduced gametes and the type of formation of the embryo and endosperm. While measuring the DNA content in the seed samples from *A. tinctoria*, only one peak was formed and clearly visible on the histograms. This suggests that only PI-stained nuclei of the embryos were detectable, and no endosperm was present. The latter was confirmed by the embryological studies on the species [65]. In this case, the observed pattern of peak formation on the histograms can be explained by two development pathways of seed formation; the fusion of reduced egg and sperm cells (amphimixis) or autonomous development of an unreduced egg cell (apomixis) (cf. [66], Figure 1). Thus, it can be inferred that flow cytometry alone cannot be used for estimating the seed development pathway in *A. tinctoria* due to the lack of endosperm in the mature seeds.

Ahmad et al. [27] established a chromosome number of 2n = 30 in the studied Greek populations of *A. tinctoria*. This number agrees with the karyological findings of other authors for the species [67,68,69], including for a Bulgarian population [70]. Ahmad et al. [27] consider *A. tinctoria* as a tetraploid species. Polyploids are known to have advantages over diploids that make them more suitable for growing as crops due to a number of gains, such as pathogen resistance, temperature stress alleviation, salinity-induced stress alleviation, and drought stress alleviation [71].

## 4. Materials and Methods

### 4.1. Plants Material

The plant material used in the study, roots, and aboveground parts were collected from plants at the flowering stage from 11 Bulgarian populations of *A. tinctoria* located in three different floristic regions of the country (Table 6, Figure 4). A commercial sample of *A. tinctoria* roots was used as a control sample (AT0).

**Figure 4 plants-12-00111-f004:**
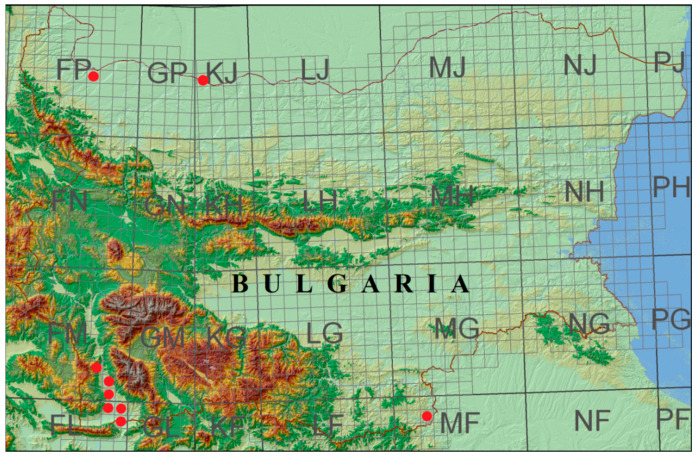
Map of the studied localities of *A. tinctoria*. ●/red circle/: the established localities.

**Table 6 plants-12-00111-t006:** List of the studied localities of *A. tinctoria* in Bulgaria.

Floristic Region	Locality	GPS Data	Altitude	Population Size, ha	Habitat
Danibian Plain	Orsoya village, Lom district (AT1)	N 43.78639 E 23.09111	30	150	Inland sandy dunes
Danibian Plain	Ostrov village, Oryahovo district (AT2)	N 43.66873 E 24.10173	30	50	Inland sandy dunes
Valley of River Struma	North of Kresna town (AT3)	N 41.73182 E 23.15583	210	0.1	Siliceous stony slope
Valley of River Struma	North of Ilindentsi village, Strumyani district (AT4)	N 41.65349 E 23.23062	380	1	Calcareous stony slope
Valley of River Struma	North of Mikrevo village, Strumyani district (AT5)	N 41.63887 E 23.17440	170	1	Siliceous stony slope
Valley of River Struma	West of Ploski village, Sandanski district (AT6)	N 41.61957 E 23.23851	300	0.1	Grassy communities on sandy soils
Valley of River Struma	North of Struma village, Sandanski district (AT7)	N 41.55849 E 23.23130	130	3.5	Grassy communities on sandy soils
Valley of River Struma	South of Lebnitsa village, Sandanski district (AT8)	N 41.50464 E 23.25049	110	0.5	Siliceous stony slope
Valley of River Struma	West of Spatovo village, Sandanski district (AT9)	N 41.50243 E 23.31010	180	2.5	Grassy communities on sandy soils
Valley of River Struma	East of Kulata village, Petrich district (AT10)	N 41.39304 E 23.37154	160	1.5	Grassy communities on sandy soils
Eastern Rhodopes Mountain	East of Odrintsi village, Ivaylovgrad district (AT11)	N 41.43415 E 26.14828	90	0.5	Grassy communities on calcareous rock and Leptosols soils

### 4.2. Methods

#### 4.2.1. Phytochemical Study

##### Extraction Procedure

One hundred (100 mg) powdered roots of each *A. tinctoria* sample were extracted with 1 mL of MeOH for 24 h at room temperature. A 50 μL (1 mg/mL) of 3,5 dichloro-4-hidroxy benzoic acid was added at the beginning of the extraction as an internal standard.

##### GC/MS Analysis

Dried methanolic extracts of the samples were dissolved in 50 μL of pyridine. Then, 50 μL of N.O-bis-(trimethylsilyl)trifluoroacetamide was added, and the samples were heated at 70 °C for 2 h. After cooling, the samples were diluted with 300 μL of chloroform and analyzed using GC-MS. The GC–MS spectra were recorded on a Thermo Scientific Focus GC coupled with Thermo Scientific DSQ mass detector (Austin, TX, USA) operating in EI mode at 70 eV. The chromatographic conditions were previously described by Berkov et al. [72]. The measured mass spectra were deconvoluted using AMDIS 2.64 software before comparison with the databases. Retention Indices (RI) of the compounds were measured with a standard n-alkane hydrocarbons calibration mixture (C9–C36) (Restek. Cat No. 31614, supplied by Teknokroma, Barcelona, Spain). The compounds were identified by comparing their mass spectra and retention indices (RI) with those of authentic standards and the National Institute of Standards and Technology (NIST) spectra library. The response ratios were calculated for each metabolite relative to the internal standard using the calculated areas for both components.

##### Spectrophotometric Analysis

The powdered roots of *A. tinctoria* samples (0.250 g) were extracted with *n*–hexane (50 mL × 1) under reflux for 20 min, and after filtration, the rest plant material was extracted with 20 mL × 2 for 15 min each extraction. The combined filtrates were poured into a 100 mL measuring flask, and it was filled to the measure with *n*-hexane. The absorption of the resulting solutions was recorded in quartz cuvettes at a length of wave λ = 520 nm against pure *n*–hexane using a spectrophotometer (Jenway 6320D). Alkannin derivatives were quantified by a standard curve, which was made by using a standard of alkannin in hexane in the concentration range of 0.02–0.7 mg/mL. Quantification was done according to Genova et al. [14]. A commercial product of *A. tinctoria* roots was used as a control sample.

#### 4.2.2. Study of Genetic Diversity

Six samples from the studied natural populations of *A. tinctoria* were included in the genetic analysis. Most of them were from Southwest Bulgaria, and one was from the Danubian plain (Table 3). A total of 2 to 12 individuals per population was studied. The samples collected were dried in silica gel before the analysis. The extraction of DNA was done using two approaches: (1) by CTAB protocol [73] and (2) by using Invisorb^®^ Spin Plant Mini Kit (Invitek Molecular GmbH, Berlin, Germany), following the instructions of the producer. The quality of the extracted DNA was tested by spectrophotometer Nanodrop^TM^ Lite (Thermo Fisher Scientific, Waltham, MA USA).

Seven Inter Simple Sequence Repeats (ISSR) markers were used to characterize the level of genetic diversity within and among populations of the species. The markers used and other details of the methods applied are presented in Table 7. The electrophoresis and other procedures were performed according to Semerdjieva et al. [25] and Petrova et al. [74].

Data analysis considered the specific patterns of ISSR markers variation. Analysis of dominant marker data, incl. ISSRs is more difficult than that of codominant data and not as straightforward. In the latter, when studying diploid organisms, heterozygotes can be scored directly from the electropherograms, while dominant markers require a different approach [75,76,77].

A binary matrix was constructed using the presence (1) and absence (0) of a particular band. Expected heterozygosity and percent of polymorphic bands were used to characterize the within-population diversity, while Nei’s [78] genetic distances among the population pairs were calculated to reveal the inter-population variation. Genetic distances were used as raw data for applying cluster analysis to reveal the grouping among the studied populations. Software DendroUPGMA (version 2002) [79] was used to construct the dendrogram reflecting the grouping of genetically similar populations.

#### 4.2.3. Flow Cytometry Analysis

Nuclear DNA content was measured by CyFlow SL Green flow cytometer (PARTEC, Germany), equipped with a green (532 nm) solid-state laser. Two types of plant material were used for the study. For precise measurement of the DNA content, fresh leaves collected in the field and then cultivated in the vegetation house of IBER—BAS plants were used for the study. *Pisum sativum* ‘Kleine Rheinländerin’ (1C = 4.38 pg, [80]) was applied as an internal standard. The plant material was treated with the extraction and staining kit ‘CyStain^®^ PI Absolute P’ (SYSMEX) following the protocol prescribed by the manufacturer. Altogether 5000 PI-stained particles were measured per run, with 5 runs per preparation. Runs with a CV above 5% for the *Alkanna* plants or the standard were discarded. For detecting any deviations in the ploidy level of the *Alkanna* individuals, mature seeds were used. Ten seeds of *Alkanna* and a piece of fresh leaf of *Pisum sativum* ‘Kleine Rheinländerin’ were co-chopped with a razor-blade and treated simultaneously with the extraction and staining kit ‘CyStain^®^ PI Absolute P’ (SYSMEX) following the protocol prescribed by the manufacturer. Only one run per preparation was measured, with 5000 PI-stained particles per run. Additionally, the same treatment of 10 seeds of *Alkanna* per preparation was done without an internal standard in order to estimate any significant deviations in the DNA content without influence from the internal standard.

## 5. Conclusions

The phytochemical analysis revealed that Bulgarian plants of *Alkanna tinctoria* (roots and aerial parts) are rich in valuable bioactive compounds such as alkannin and ononitol, especially those from Kresna (AT3), North of Ilindentsi village (AT4), South of Lebnitsa village (AT8), in the Struma valley, Ostrov village (AT2), in the Danubian Plain. Plant material from these locations was recommended as a promising source for cultivation as a crop in order to provide a raw material for the established market needs and to preserve the natural populations of the species. However, further agronomic studies would need to be conducted to identify the optimal soil, water, nutrient requirements, harvest, and post-harvest handling of *Alkanna tinctoria* as an agronomic crop, which may take a significant amount of time.

The present study demonstrated that aerial parts of the plant are also promising as a source of valuable biologically active substances such as ononitol, which has shown pronounced hepatoprotective, analgesic, and anti-inflammatory activity. This study reported ononitol as a newly discovered inositol for the genus *Alkanna* and the Boraginaceae family.

## Figures and Tables

**Figure 1 plants-12-00111-f001:**
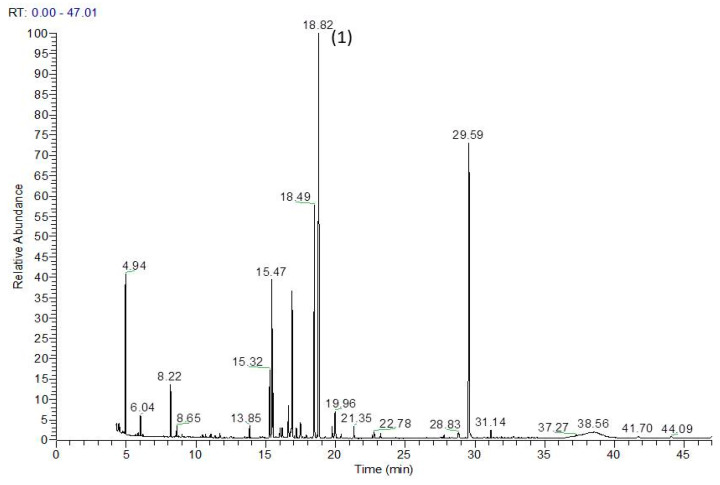
Chromatogram representative of methanolic extracted from *A. tinctoria* aerial parts. Peak 1: ononitol.

**Figure 2 plants-12-00111-f002:**
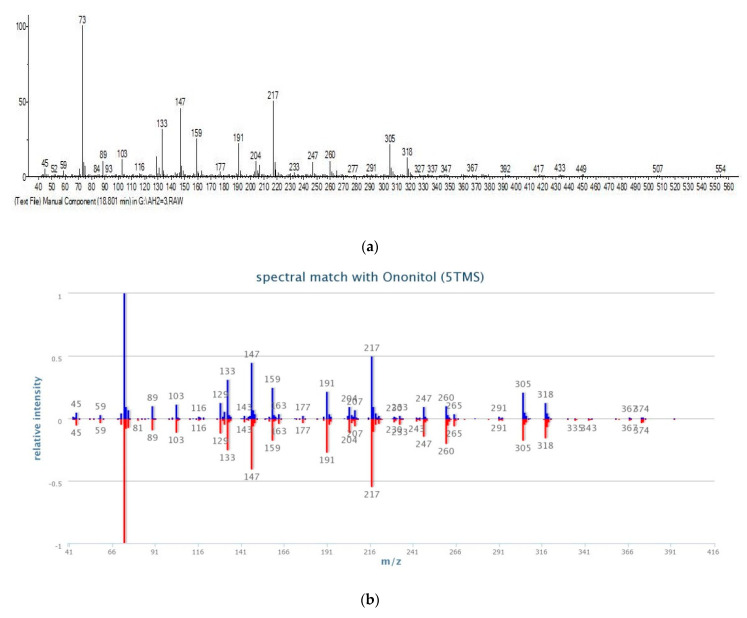
GC/MS spectra of (**a**) derivatized ononitol of methanolic extract, and (**b**) ononitol from the extract and MS spectra of standard of Golm Metabolome Database (GMD).

**Figure 3 plants-12-00111-f003:**
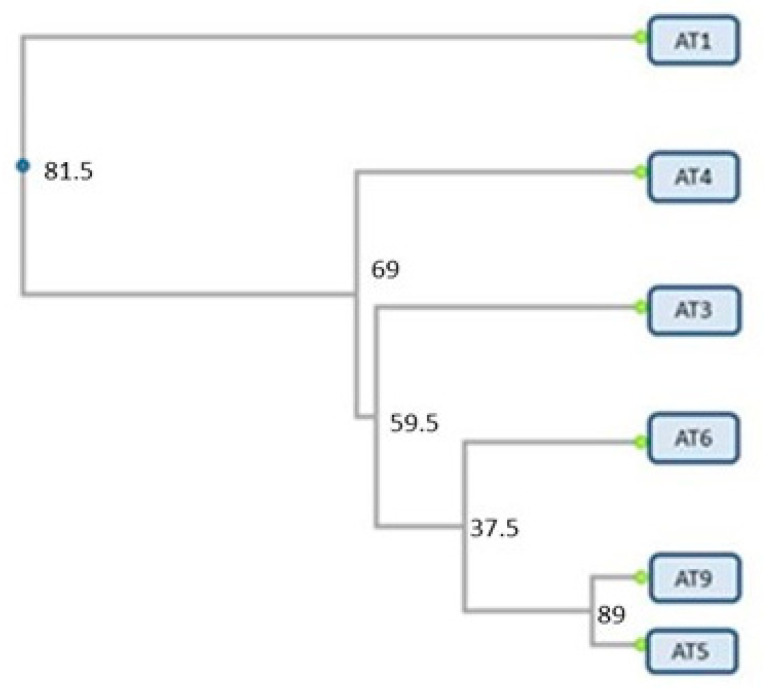
Cluster dendrogram based on the Nei’s genetic distances among populations. Codes of the samples (AT1, AT3, AT4, AT5, AT6, AT9) are given according to Table 6.

**Table 1 plants-12-00111-t001:** Comparative summary data on metabolite profile of the aerial parts and roots of the studied *A. tinctoria* populations.

Identified Compounds	Roots *	Aerial Parts *
**Phenolic acids**		
4-Hydroxybenzoic acid	1.4	4.7
Vanilic acid	0.4	0.3
Protocatechuic acid	1.0	1.3
Ferulic acid	0.5	1.1
Caffeic acid	11.5	38.1
**Organic acids**		
Succinic acid	10.8	65.5
Glyceric acid		5.9
Malic acid	8.0	12.9
**Sugars and polyols**		
Fructose 1	163.9	106.7
Fructose 2	226.8	56.3
Glucose	281.2	137.6
Ononitol		2313.1
Myo-Inositol	60.7	27.8
Sucrose	2272.1	1232.1
**Lipids**		
Hexadecanoic acid	45.1	70.4
Octadecanoic acid	12.4	18.2
Teracosanol	5.0	1.9
Hexacosanol	3.7	
Campestrol	8.1	4.6
Stigmasterol	3.1	
β-Sitosterol	43.3	24.9

* The presented values are mean of all studied populations. The amounts of metabolites are represented as response ratio that represents peak area ratios using 3,5 dichloro-4-hidroxybenzoic acid (50 μg) as quantitative internal standard.

**Table 2 plants-12-00111-t002:** Spectrophotometric determination on the alkannin content in the studied samples of *Alcanna tinctoria* roots.

Sample ^1^	Alkannin Content to Air Dry Weight, g [%]
AT0	4.4 ± 0.1
AT1	2.4 ± 0.1
AT2	3.8 ± 0.1
AT3	5.9 ± 0.1
AT4	3.6 ± 0.1
AT5	2.9 ± 0.1
AT6	1.9 ± 0.1
AT7	1.8 ± 0.5
AT8	4.2 ± 0.9
AT9	1.4 ± 0.4
AT10	3.2 ± 0.2
AT11	1.2 ± 0.1

^1^ Codes of the samples (AT1–AT11) are given according to Table 6 (Section 4). AT0—a control (commercial product).

**Table 3 plants-12-00111-t003:** Natural populations included in the genetic studies and polymorphism and diversity.

Population (Abbreviation)	Sample Size	Percent of Polymorphic Bands	Gene Diversity He ± SD
AT5	8	69.1	0.236 ± 0.110
AT1	12	72.1	0.228 ± 0.056
AT3	7	59.3	0.298 ± 0.060
AT6	2	55.3	0.220 ± 0.110
AT4	8	58.2	0.248 ± 0.096
AT9	2	55.3	0.224 ± 0.132
Mean	6.5	61.5	0.242

**Table 4 plants-12-00111-t004:** DNA content in fresh leaves.

Sample	1C [pg] 1st Run	1C [pg] 2nd Run	1C [pg] 3rd Run	1C [pg] 4th Run	1C [pg] 5th Run	1C [pg] Mean	CV [%]
AT 5	1.333	1.327	1.323	1.324	1.324	1.326	0.33
AT 7	1.326	1.326	1.325	-	-	1.326	0.02

**Table 5 plants-12-00111-t005:** DNA content in the seeds of the studied population of *A. tinctoria*.

Sample	No. of Seeds Per Run	1C [pg]	CV *Alkanna* [%]
AT1	10	1.29	6.6
AT2	10	1.31	6.1
AT3	10	1.43	7.5
AT4	10	1.42	8.4
AT5	10	1.28	5.8
AT6	10	1.39	5.3
AT7	10	1.33	4.7
AT8	10	1.43	8.5
AT9	10	1.30	5.1
AT10	10	1.32	8.0
AT11	10	1.49	10.7

**Table 7 plants-12-00111-t007:** ISSR primers studied for the study of genetic diversity of *A. tinctoria*.

ISSR Primer (Name and Sequence)		Optimal Annealing Temperature	Total Number of Bands	Number of Polymorphic Bands
UBC-807	5′-AGAGAGAGAGAGAGAGT-3′	59	11	8
UBC-811	5′-GAGAGAGAGAGAGAGAC-3′	53	12	9
UBC-827	5′-ACACACACACACACACG-3′	57	8	5
UBC-835	5′-AGAGAGAGAGAGAGAGYC-3′	54	9	6
UBC-845	5′-CTCTCTCT CTCTCTCTRG-3′	54	13	9
UBC-846	5′-CACACACACACACACAAGT-3′	54	10	8
UBC-856	5′-ACACACACACACACACCTA-3′	54	11	8

## Data Availability

Not applicable.

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
