# Peer review of "Metabolic Profiles, Genetic Diversity, and Genome Size of Bulgarian Population of Alkanna tinctoria"

_plants, 2022, doi:10.3390/plants12010111_

Round 1
Reviewer 1 Report
A Table is required summarizing all the chemical structures and composition of plant's organs by area.
Author Response
Dear reviewer
The authors are thankful for the constructive comments and suggestions.
Below, please, find “point by point” answers to your comments
A Table is required summarizing all the chemical structures and composition of plant's organs by area.
Answer: We added a table in the text (Table 1) with summarized comparative data of qualitative and quantitative composition of the biologically active compounds per plant parts (roots and aerial parts) and per populations.
Reviewer 2 Report
please see attachment

Author Response
Dear reviewer
The authors are thankful for the constructive comments and suggestions.
Below, please, find “point by point” answers to your comments
The manuscript deals with “Metabolic profiles, genetic diversity and genome size of Bulgarian population of Alkanna tinctoria”, a medicinally important plant that produces several biologically active naphthoquinones and many secondary products of interest. The authors identified several primary and secondary compounds that are of general interest. In particular, the authors claim that ononitol, a sugar alcohol has been identified for the first time in the Boraginaceae family (Alkanna tinctoria). The metabolites have been identified using GC-MS analysis. The data are interesting and warrant publication.
However, following are minor corrections that need the attention of the authors.
Page 1, Introduction: Second paragraph: The sentence should read as “As the root is the used plant part” (not plan part).
Answer: Corrected as suggested
Page 2: First line: “It should read as biologically active compounds”.
Answer: Corrected as suggested
Page 2: Second para: First line: It should be “a” (not “an”).
Answer: Corrected as suggested
Page 2. Para 4. It is useful (not usefull).
Answer: Corrected as suggested
Page 7. Para 1, Line 3: Put full point after 4.7-10.7%.
Answer: Corrected as suggested
Page 7. First para: It should be previously been focused on A. tinctoria.
The sentence should read as “drought and salts [44, 45]. (delete (.
Also after [46]. Delete (.
Answer: Corrected as suggested
Page 7. 3.1.2. It should read as “strongly affected by light” (not effected).
Answer: Corrected as suggested
Page 8. Second paragraph, first line: The sentence should read as “It is demonstrated in this study that the low level of……”
Answer: Corrected as suggested
Delete [Hutch and Hich 1999] after the reference number 49.
Answer: Corrected as suggested
Delete [Wang 2020] after the reference number 50.
Answer: Corrected as suggested
Reference 27: Sometimes it is written as Ahmat (27), and other times as Ahmad (27). Which one is correct?
Answer: Thank you for noticing it. The correct name is Ahmad and it was corrected throughout the manuscript.
Reviewer 3 Report
The MS well prepared; Authors himself identified what they need to include for making the study more significant. Authors have stated that in "However, further agronomic studies would need to be conducted to identify the optimal soil, water, nutrient requirements, harvest and post-harvest handling of A. tinctoria as agronomic crop" These are the factor which are responsible for the varying amount of Alkannin etc. in plants of different locality. so it is important to discuss these too. and this should be the main focus of study.
Author Response
Dear reviewer
The authors are thankful for the constructive comments and suggestions.
Below, please, find “point by point” answers to your comments
The MS well prepared; Authors himself identified what they need to include for making the study more significant. Authors have stated that in "However, further agronomic studies would need to be conducted to identify the optimal soil, water, nutrient requirements, harvest and post-harvest handling of A. tinctoria as agronomic crop" These are the factor which are responsible for the varying amount of Alkannin etc. in plants of different locality. so it is important to discuss these too. and this should be the main focus of study.
Answer: The conducted research is part of a project aimed at establishing the optimal conditions for the growth of Alkanna tinctoria as an agricultural crop. The first step in this process is the identification of the qualitative and quantitative composition of the biologically active compounds (mainly the content of alkannin) in the different natural populations of the species, in order to select accessions with desirable traits. The second step that may take few years is to conduct comparative study on productivity of these accessions grown side by side